# Dietary Supplementation with Prebiotic Chitooligosaccharides Enhances the Growth Performance, Innate Immunity and Disease Resistance of Nile Tilapia (*Oreochromis niloticus*)

Nurmalasari [1,2,†], Chun-Hung Liu [3,4,†], Ir. M. Maftuch [2] and Shao-Yang Hu [1,4,*]

1   Department of Biological Science and Technology, National Pingtung University of Science and Technology, Pingtung 912, Taiwan
2   Department of Fisheries and Marine Science, University of Brawijaya, Malang 65145, Indonesia
3   Department of Aquaculture, National Pingtung University of Science and Technology, Pingtung 912, Taiwan
4   Research Center for Animal Biologics, National Pingtung University of Science and Technology, Pingtung 912, Taiwan
*   Correspondence: syhu@mail.npust.edu.tw; Tel.: +886-8-7703202 (ext. 6356); Fax: +886-8-77405584
†   These authors contributed equally to this work.

**Abstract:** Prebiotics acting as immunosaccharides exhibit immunomodulatory functions to improve the immune defense of the host against infectious diseases. The purpose of the present study was to investigate the effects of dietary chitooligosaccharide (COS) supplementation on the growth, innate immunity and disease resistance of Nile tilapia (*Oreochromis niloticus*) fed a diet containing 0.4%, 0.8% and 1.2% COSs for 8 weeks. The results showed significant increases in weight gain (WG), feed efficiency (FE) and specific growth rate (SGR) in COS-supplemented fish compared to fish in the control group. The fish exhibited a significant decrease in cumulative mortality in fish fed 0.8% and 1.2% COS when challenged with *Streptococcus iniae*. The immune parameters, including phagocytotic activity (PA), respiratory burst (RB) activity, and superoxide dismutase (SOD) activities of the head kidney and serum lysozyme, as well as the expression of *TNF-α*, *IL-1β* and *IL-8*, were revealed in 0.8% and 1.2% COS-supplemented fish. These results demonstrated that COS could be used as a prebiotic and that dietary supplementation with 0.8% COS could improve growth performance and innate immunity against pathogen infections in Nile tilapia.

**Keywords:** chitooligosaccharides; *Oreochromis niloticus*; growth; innate immunity; disease resistance





## 1. Introduction

Tilapia is one of the most important cultured commercial fish species worldwide due to its properties of rapid growth, high tolerance to environmental stress, desirable taste and high marketability. This species has been farmed in over 135 countries and serves as an important source of animal protein and income in developing countries. According to statistical data from the Food and Agriculture Organization (FAO, 2022), the global production yield of tilapia was around 5.47 million tons, which accounted for 11.2% of global finfish production in aquaculture, in 2020 [1]. However, an intensive culture for perusing high production usually causes deterioration of the culture environment, which results in a high incidence of bacteria-induced diseases and jeopardizes the sustainable development of the tilapia aquaculture industry. For example, streptococcosis caused by the pathogen *Streptococcus* species often results in hemorrhagic septicemia and severe mortality in tilapia [2,3]. Facing disease problems, antibiotics are commonly used in fish farming for prophylactic or therapeutic purposes to prevent disease outbreaks. However, the increasing global concerns about antibiotic issues, such as the spread of drug-resistant pathogens, impacts of microbial ecology and risk of food safety hazards, have prompted research to develop alternative immunostimulants for improving immunity against diseases in aquaculture.

Prebiotics are defined as natural substances in food that contribute beneficial effects to the host by enriching growth or activity of a limited number of beneficial bacteria in the gut. Carbohydrate compounds or derivatives have been considered one type of prebiotic, and their efficacy in host health is the most extensively investigated. The mechanisms by which prebiotics contribute benefits to the host are thought to involve carbohydrates fermented by specific probiotics in the gut, and the production of metabolites, such as short-chain fatty acids (SCFAs), lactate, pyruvate and succinate, can act as an energy source for colonocytes and as immunostimulants to inhibit the growth of pathogenic organisms and improve immunity by activating immune cells [4]. Moreover, the presence of prebiotics, such as oligosaccharides, in the gut has been shown to increase [5].

Chitin, which exists in nature as ordered microfibers, is a long-chain polysaccharide polymer of N-acetylglucosamine. It is the second most abundant natural biopolymer and is widely distributed in the exoskeletons of marine crustaceans, crabs and shrimp, as well as the cell walls of fungi and yeast. For biomedical and agricultural applications, chitin is usually converted to its deacetylated derivative, chitosan, which is a polysaccharide composed of β-(1,4)-linked D-glucosamine. However, chitin and chitosan exhibit the drawback of low solubility under physiological conditions for biological utilization. Chitooligosaccharides (COSs) are degraded derivatives of chitin or chitosan from the enzymatic or chemical hydrolysis of chitosan, resulting in an oligo mixture with 3–10 saccharide residues. Investigations have shown that the biological activities of COSs are strongly associated with their structural features, such as degree of polymerization and acetylation, pattern of acetylation and electronic distribution [6,7]. The lower degree of polymerization and the lower molecular weight of COSs are the lower the viscosity, and these COSs can be easily and effectively absorbed by the intestine, therefore enabling higher biological activities [8]. Thus, different approaches have been developed to prepare COSs with low molecular weights [9]. Recently, COSs have attracted increasing attention in terms of their biomedical or agricultural applications due to their relatively low molecular weight, nontoxicity, high solubility, biocompatibility, biodegradability and beneficial effects on hosts. For instance, $Fe_3O_4$ nanoparticles coated with COSs exhibiting higher cytocompatibility have been developed as potential materials in drug delivery systems, cell immobilization and cell tracking for various biomedical applications [10]. In the livestock and poultry industry, dietary supplementation with COSs has been shown to improve growth performance and attenuate enterotoxigenic *Escherichia coli*-challenged intestinal inflammation in piglets [11]; in ovo supplementation with COSs can modulate the health of gut microbiota in broiler chickens [12]. In aquaculture, reports have also shown that dietary COS supplementation improves innate immunity against pathogen infections in diverse fish species, including olive flounder (*Paralichthys olivaceus*) [13], striped catfish (*Pangasianodon hypophthalmus*) [14], ovate pompan (*Trachinotus ovatus*) [15], and blunt snout bream (*Megalobrama amblycephala*) [16]. These studies elucidated that prebiotic COS supplementation is a potential immunostimulant that can strengthen defense against infectious diseases in aquaculture.

In the present study, the efficacy of dietary COS on growth performance and nutrient utilization was assayed by evaluating growth parameters such as weight gain (WE), feed efficiency (FE) and mRNA expression of nutrient metabolism and growth-related genes. Innate immune parameters, including phagocytic activity (PA), respiratory burst (RB), superoxide dismutase (SOD) activity in head kidney leukocytes and serum lysozyme (LYZ), and the expression of cytokine genes were used to evaluate the efficacy of COS on the health status of tilapia. The efficacy of COS on disease resistance against pathogen infection was evaluated by challenge with *Streptococcus iniae*. These results could provide vital information in reference to the feasibility of COSs as an immunostimulant in aquaculture.

## 2. Materials and Methods

### 2.1. Fish Husbandry and Bacterial Strain

Juvenile Nile tilapia (*Oreochromis niloticus*) were obtained from fish farm located at Pingtung city, Taiwan. The fish were acclimatized in 150-L glass tank with 120 L of

filtered and aerated freshwater at 28 °C for 7 days in the Aquatic Laboratory of National Pingtung University (NPUST). The experiments were conducted in compliance with local animal welfare regulations and approved by the NPUST Institutional Animal Care and Use Committee (approval no. NPUST-107-063). The cultivation of pathogen *Streptococcus iniae*, which isolated from diseased tilapia, was described in a previous report [17].

### 2.2. Preparation of the Experimental Diet and Feeding Trial

COSs with an average molecular weight of 800 Da were purchased from Everest Food Ingredients Laboratories Corp., Taichung, Taiwan. Different amounts of COSs were added to the basal diet, resulting in a diet containing 4, 8, or 12 g/kg COSs. Formulations and proximate compositions of the experimental diets are listed in Table 1. The experimental diets were prepared by a modified protocol in accordance with a previous report [17]. Briefly, feed stuffs were ground in a hammer mill to pass through an 80-mesh screen (150 μm). The ingredients were thoroughly mixed in a mixer until a stiff dough resulted. Dough was passed through a paste extruder, resulting in feed pellets with particle sizes of approximately 1~5 mm. The feed pellets were dried in a drying cabin at 30 °C to a moisture level less than 12%. The dry pellets were preserved in plastic bins at 4 °C until use. Juvenile tilapia (1.04 ± 0.3 g) were divided into four groups and randomly distributed into 12 glass tanks at a density of 30 fish per tank. Each group was carried out in triplicate in a tank (85 cm × 40 cm × 45 cm) with an independent recirculation system. The fish in the control, G1, G2 and G3 groups were fed a basal diet and basal diet containing 4, 8, and 12 g/kg COS, respectively. The fish were fed twice daily at 5% of body weight for 8 weeks. The fish were weighed individually once every two weeks, and the feed supplementations were adjusted based on the mean weights o the fish. Uneaten feeds were collected after 1 h feeding, and the fecal matters were separated from uneaten feed by using a drooping pipette. Then, uneaten feeds were dried in an oven at 80 °C to determine feed intake. The parameters involved in growth and innate immunity, digestive enzyme activities, and mRNA expression of indicator genes were analyzed, and the challenging test conducted at the end of the feeding trial.

**Table 1.** Ingredients and proximate composition of the experimental diets.

| Ingredients | Administrations | | | |
|---|---|---|---|---|
| | Control | G1 | G2 | G3 |
| COSs | 0 | 4 | 8 | 12 |
| Fish meal | 50 | 50 | 50 | 50 |
| Soybean meal | 300 | 300 | 300 | 300 |
| Wheat middling | 161 | 161 | 161 | 161 |
| Rice bran | 300 | 300 | 300 | 300 |
| α-starch | 60 | 60 | 60 | 60 |
| Cellulose | 70 | 66 | 62 | 58 |
| Skim milk | 10 | 10 | 10 | 10 |
| Soybean oil | 30 | 30 | 30 | 30 |
| Mineral mixture [a] | 16 | 16 | 16 | 16 |
| Vitamin mixture [a] | 3 | 3 | 3 | 3 |
| Proximate composition [b] | | | | |
| Crude protein (%) | 26.8 | 27.3 | 27.4 | 27.5 |
| Crude lipid (%) | 12.1 | 12.2 | 12.6 | 12.7 |
| Moisture (%) | 10.4 | 10.4 | 10.6 | 10.5 |
| Ash (%) | 15.0 | 15.1 | 15.6 | 15.7 |

[a] Mineral mixture (mg·kg$^{-1}$ of mixture) and vitamin mixture (mg·kg$^{-1}$ of mixture) purchased from the Shinta Food Company, Pingtung. [b] The proximate compositions of the experimental diet were determined by AOAC method [18].

*2.3. Growth Parameters*

The body weights (BWs) of all tilapia from each tank were measured every two weeks. At the termination of the feeding trial, the growth performance parameters, including weight gain (WG), specific growth rate (SGR), feed efficiency (FE) and survival rate (SR), were calculated using the following equations: WG = final BW-initial BW; SGR = (final BW-initial BW) $\times$ 100/duration of the experiment in days; FE = (final BW-initial BW)/feed intake; SR = 100 $\times$ (final number of test fish)/(initial number of test fish).

*2.4. Determination of Digestive Enzyme Activities in the Gut*

Six fish from each group were sacrificed and dissected at the end of the feeding trial. The protocol for preparing extracted suspension from tilapia intestine and digestive protease, amylase, cellulase and xylanase activities of intestinal extracts were carried out in accordance with a previous report [19]. Digestive lipase activity was assayed spectrophotometrically by hydrolysis of 4-nitrophenyl butyrate (4-NPB) using a modified protocol from a previous report [20]. Briefly, 100 μL of supernatant was mixed with 100 μL of phosphate buffered saline (PBS) containing 0.5% Triton X-100 and 0.5 mM 4-NPB solution and incubated at 37 °C for 30 min. 4-Nitrophenol (4-NP) was used as a standard to establish a calibration curve. 4-NP released from 4-NPB by lipase activity was measured at 405 nm with a spectrophotometer. One unit of lipase activity was defined as the amount of enzyme that released 1 μmole of 4-NP per minute. The total protein of the suspension was measured by the Bradford method [21].

*2.5. Evaluation of Innate Immune Parameters*

At the termination of the feed trial, six fish from each group were sampled to evaluate the efficacy of COS on innate immunity by determining the phagocytic activity (PA), respiratory burst (RB) activity and superoxide dismutase (SOD) activity of the head kidney and lysozyme activity of serum. The protocol for isolating leukocytes of head kidney and serum and for analyzing PA and RB activities were performed following a previously reported method [22].

2.5.1. Phagocytic Activity Assay

Briefly, $10^6$ leukocytes/well were cultured in a 24-well microplate containing 300 μL/well of L-15 medium at 28 °C for 1 hour. Then, 300 μL of L-15 medium containing fluorescent latex beads was added to each well and incubated at 28 °C for 2 h in the dark. The wells were then rinsed with PBS two times to remove unengulfed fluorescent latex beads, and the cells were then fixed with 300 μL of 1% formalin for 30 min in the dark. The fixed cells were stained with propidium iodide for 10 min and washed twice again with PBS. Three hundred phagocytes were analyzed via fluorescence microscopy (Olympus IX 50, Tokyo, Japan). The ratio of leukocytes that engulfed one or more fluorescent latex beads, defined as the PA, was expressed as follows: PA (%) = [100 $\times$ (number of phagocytic leukocytes with engulfed fluorescent latex beads) $\times$ (number of total leukocytes)$^{-1}$].

2.5.2. Respiratory Burst Activity Assay

Leukocytes from the head kidney were inoculated into a 96-well microplate to a concentration of $10^6$ cells/well and cultured in 0.1 mL of L-15 medium at 28 °C for 30 min. Then, 0.1 mL of 0.1% zymosan in modified complete Hank's balanced salt solution (McHBSS) was added to the wells and incubated at 28 °C for 30 min to produce superoxide. Subsequently, the supernatants were removed, and the leukocytes were washed twice with McHBSS and then stained with 100 μL of a 0.3% nitro blue tetrazolium (NBT) solution at 28 °C for 30 min. The reaction was stopped by adding 100 μL of 100% methanol, and then the cells were washed three times with 70% methanol. Formazan was dissolved by adding 120 μL of 2 M KOH and 140 μL of DMSO for 2 min. Respiratory burst activity based on NBT reduction was determined by measuring at an absorbance of 630 nm.

### 2.5.3. SOD and Lysozyme Activity Assay

SOD activity was measured based on its ability to inhibit the photoreduction of NBT reactions as described previously [23]. Briefly, $10^6$ leukocytes from head kidney were homogenized in 0.5 mL of PBS (pH 7.4) and centrifuged at $10,000 \times g$ at 4 °C for 15 min. The supernatant as crude enzyme was used for the SOD activity assay. The reaction mixture in a centrifuge tube containing 350 μL of the mixture, which comprised 100 μL of crude enzyme, 62.5 μL of 150 mM PBS buffer, 37.5 μL of 130 mM methionine, 37.5 μL of 1 mM EDTA, 37.5 μL of 0.63 mM NBT and 75 μL of 75 mM riboflavin, was reacted at 28 °C for 10 min. One unit of SOD activity was defined as the amount of enzyme necessary to produce a 50% inhibition of the NBT reduction rate measured at 530 nm using a microplate reader. SOD activity was expressed as U/mg protein. A turbidimetric assay was used to measure serum lysozyme activity, in which chicken egg white lysozyme (Sigma-Aldrich, St. Louis, MO, USA) lyse *Micrococcus lysodeikticus* was a standard, as described previously [24]. A mixture consisting of 10 mL of serum sample or different concentrations of chicken egg white lysozyme and 200 mL of *Micrococcus lysodeikticus* (0.2 mg/mL) in 50 mM PBS buffer (pH 6.2) was reacted in a 96-well microplate. The reaction was performed at 25 °C, and the absorbance was measured at 450 nm using a microplate reader after 1 and 6 min. One unit was defined as the amount of sample required to reduce absorbance by 0.001 min$^{-1}$ at 450 nm. The concentration of protein in the leukocyte suspension and serum sample were determined via the Bradford method using bovine serum albumin as a standard with the Bio-Rad Protein assay reagent (Bio-Rad Laboratories, Mississauga, ON, Canada).

### 2.6. Determination of Indicator Gene Expression

Total RNAs were extracted from liver, head kidney and spleen of fish in control, G1, G2 and G3 groups using the TriPure isolation reagent (Roche, Mannheim, Germany) according to the manufacturer's protocol. The cDNA was synthesized from 1 μg of total RNA using an iScript cDNA Synthesis Kit (Bio-Rad, Hercules, CA, USA). The mRNA expression levels of indicator genes were determined by quantitative real-time PCR (qRT-PCR). The expression of β-actin was used as an internal control and all primers designed for the target genes are listed in Table 2. Each qRT-PCR was conducted in a mixture containing 1 μL cDNA template, 1 μL each of forward and reverse primer, 10 μL KAPA SYBR FAST qPCR Master Mix (KAPA KR0389, Wilmington, MA, USA) and 8 μL ddH$_2$O to make up the total volume of 20 μL. qRT-PCR was performed using StepOnePlus Real-Time PCR system (Applied Biosystems, Waltham, MA, USA) under the following program: 60 °C for 2 min; 95 °C for 10 min followed by 40 cycles of 95 °C for 15 s, 60 °C for 1 min, and 60 °C for 1 min, and finally at 4 °C for 5 min. The relative expression level of the target genes normalized to β-actin is expressed as the mean ± standard error (SE).

### 2.7. Challenge Test

To investigate the resistance of Nile tilapia to the pathogen *S. iniae*, ten fish from the control, G1, G2 and G3 groups were intraperitoneally (I.P.) injected with 50 μL of saline containing *S. iniae* at the respective 50% lethal dose (LD$_{50}$) concentration of $10^5$ CFU per fish. The challenge test was carried out in triplicate, and experimental fish were maintained in a tank containing 40 L of fresh water at 28 ± 1 °C. Fish fed the control diet and injected with saline or *S. iniae* were used as the positive and negative controls, respectively. The challenged fish were checked daily, and dead fish were removed slowly from aquariums to avoid stress. Cumulative mortalities were recorded in all the groups for 7 days post infection.

### 2.8. Statistical Analysis

Experimental values were analyzed using one-way analysis of variance (ANOVA). Tukey's multiple comparison test to examine significant differences among treatments. Cumulative mortality in the challenged tested was analyzed by the Kaplan-Meier method.

Statistical analysis of data among the treatments was performed using SAS software (SAS Institute, Cary, NC, USA). The value of $p < 0.05$ was statistically significant.

**Table 2.** Primers used for detecting expression level of specific genes in the present study.

| Gene Name | Primer Sequences (5′→3′) | Amplicon Size | Accession No. |
|---|---|---|---|
| Glucose kinase (*GK*) | F: GCAGCGAGGAAGCCATGAAGA<br>R: GAGGTCCCTGACGACTTTGTGG | 101 bp | XM_003451020 |
| Glucose-6-phosphatase (*G6Pase*) | F: AGCGCGAGCCTGAAGAAGTACT<br>R: ATGGTCCACAGCAGGTCCACAT | 107 bp | XM_003448671 |
| Growth hormone receptor-1 (*GHR-1*) | F: GAATACAAGTCCTTCCGGGCTAA<br>R: CTCATACTCCACACGCATCCA | 100 bp | AY973232 |
| Insulin-like growth factor-1 (*IGF-1*) | F: TGTCTGCCAGTAAGGATGTTCTTG<br>R: GGCTTTCCACGCCACTTAAC | 100 bp | EU272149 |
| Tumor necrosis factor-α (*TNF-α*) | F:CCAGAAGCACTAAAGGCGAAGA<br>R:CCTTGGCTTTGCTGCTGATC | 82 bp | AY428948 |
| Interleukin-1β (*IL-1β*) | F: TGTCGCTCTGGGCATCAA<br>R: GGCTTGTCGTCATCCTTGTGA | 63 bp | KJ574402 |
| Interleukin-8 (*IL-8*) | F: CCTCGAGAAGGTGGATGTGAA<br>R: CATGAGACCCAGGGCATCA | 100 bp | GQ355864 |
| β-actin | F: CCACACAGTGCCCATCTACGA<br>R: CCACGCTCTGTCAGGATCTTCA | 111 bp | EU887951 |

## 3. Results

### 3.1. COS Supplementation Enhances Growth Performance

The effects of COSs on growth performance were assessed by evaluating parameters, including weight gain (WG), feed efficiency (FE) and specific growth rate (SGR). The growth performance and feed utilization of Nile tilapia after 8 weeks of COS feeding are shown in Table 3. There was no significant difference in the survival rate between the groups, suggesting that supplementation with COSs in each group was harmless to fish. The WG, FE and SGR in COS-supplemented tilapia of the G1, G2 and G3 groups were significantly increased compared to those in tilapia of the control group. The efficacy of COS supplementation in WG and SGR was significantly accompanied by an increase in COS levels in the diet (G3 > G2 > G1). Moreover, there was no significant difference in FE between the G2 and G3 groups, suggesting that supplementation with 0.8% COSs was sufficient to improve FE.

**Table 3.** Growth performance and feed utilization of Nile tilapia (*O. niloticus*) after feeding control diet and control diet containing 4, 8 and 12 g kg$^{-1}$ of COS for 8 weeks.

| Parameters | Administrations (COS Supplementation Level g kg$^{-1}$) | | | |
|---|---|---|---|---|
| | **0 (Control)** | **4 (G1)** | **8 (G2)** | **12 (G3)** |
| Initial weight (g) | 1.04 ± 0.02 [a] | 1.04 ± 0.04 [a] | 1.04 ± 0.09 [a] | 1.04 ± 0.04 [a] |
| Finial weight (g) | 10.55 ± 0.9 [d] | 14.09 ± 0.97 [c] | 15.50 ± 0.51 [b] | 17.49 ± 0.37 [a] |
| Weight gain (WG) (g) | 9.50 ± 0.1 [d] | 13.86 ± 0.48 [c] | 15.23 ± 0.51 [b] | 16.45 ± 0.54 [a] |
| Feed efficiency (FE) | 0.66 ± 0.01 [d] | 0.72 ± 0.03 [c] | 0.76 ± 0.03 [ab] | 0.78 ± 0.04 [a] |
| Specific growth rate (SGR) | 1.51 ± 0.07 [d] | 2.09 ± 0.08 [c] | 2.31 ± 0.09 [b] | 2.51 ± 0.07 [a] |
| Survival rate (%) | 94.4 ± 1.9 [a] | 94.4 ± 1.9 [a] | 97.7 ± 3.8 [a] | 97.7 ± 3.8 [a] |

Data are presented as the mean ± S.E. from triplicates of each group. Different superscripts in the same rows represent significant differences ($p < 0.05$)

*3.2. COS Supplementation Enhances Digestive Enzyme Activity and Gene Expressions Involved in Nutrient Metabolism*

The improved WG, FE and SGR in COS-supplemented tilapia urged us to further investigate the digestive enzyme activities in the fish gut and nutrient mentalism at the molecular level in the liver. As shown in Table 4, although there was no significant difference in xylanase activity in tilapia between groups, intestinal protease, amylase, cellulase and lipase activities were significantly increased in fish of the G2 and G3 groups compared to those in fish of the control group. Moreover, the protease activity of tilapia in the G1 group was also obviously higher than that of fish in the control. The intestinal protease and amylase activity of tilapia was significantly increased in the G3 group compared with the G1 and G2 groups. Significant increases in hepatic *GK*, *G6Pase*, *GHR* and *IGF-1* gene expression were observed in tilapia from the G2 and G3 groups compared to those in fish from the control group. Moreover, there was a significant increase in *GK* and *G6Pase* mRNA expression in tilapia from the G2 and G3 groups compared to tilapia from the G1 group. There were no significant differences in hepatic *GK*, *G6Pase*, *GHR* and *IGF-1* expression in tilapia between the G2 and G3 groups or between the control and G1 groups (Figure 1). The increased mRNA expression of glucose metabolism- and growth-related genes in tilapia of the G2 and G3 groups supports the results of the feed efficiency and growth enhancement in tilapia supplemented with 0.8% and 1.2% COSs in the feeding trial.

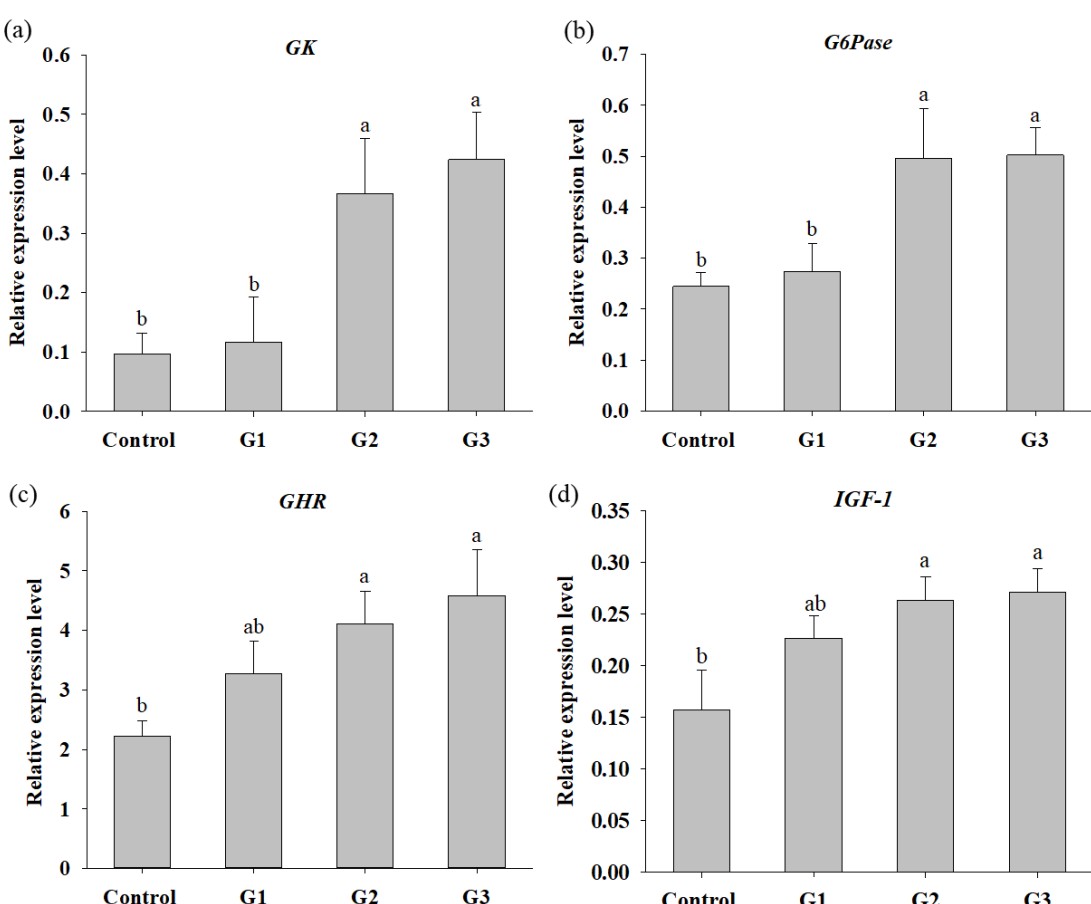

**Figure 1.** Relative expression levels of hepatic genes participated in glucose metabolism and growth in Nile tilapia fed with control diet and control diet containing 4 (G1), 8 (G2) or 12 (G3) g/kg COS for 8 weeks: (**a**). glucokinase (GK); (**b**) glucose-6 phosphatase (G6Pase); (**c**) growth hormone receptor (GHR); (**d**) insulin-like growth factor (IGF-1). Bars with different superscripts are significantly different ($p < 0.05$).

**Table 4.** Intestinal digestive enzymes of Nile tilapia (*O. niloticus*) after feeding basal diet and basal diet containing 4, 8 and 12 g kg$^{-1}$ of COS for 8 weeks.

| Activity (U/mg) | Control | G1 | G2 | G3 |
|---|---|---|---|---|
| Protease | 0.23 ± 0.08 [c] | 0.36 ± 0.07 [b] | 0.39 ± 0.04 [b] | 0.54 ± 0.09 [a] |
| Amylase | 0.27 ± 0.06 [c] | 0.31 ± 0.04 [bc] | 0.35 ± 0.02 [b] | 0.49 ± 0.02 [a] |
| Cellulase | 0.23 ± 0.05 [c] | 0.35 ± 0.09 [bc] | 0.41 ± 0.07 [ab] | 0.58 ± 0.09 [a] |
| Xylanase | 0.17 ± 0.03 [a] | 0.16 ± 0.03 [a] | 0.21 ± 0.07 [a] | 0.18 ± 0.05 [a] |
| Lipase | 0.07 ± 0.008 [b] | 0.09 ± 0.01 [b] | 0.16 ± 0.04 [a] | 0.21 ± 0.07 [a] |

Data are presented as the mean ± S.E. from triplicates of each group. Different superscripts in the same rows represent significant differences ($p < 0.05$).

### 3.3. COS Supplementation Enhances Innate Immunity

The immune parameters, including the phagocytic activity (PA), respiratory burst (RB) activity and superoxidase dismutase (SOD) activity of head kidney leukocytes and serum lysozyme activity, were evaluated after COS supplementation for 8 weeks. The PA, RB and SOD activities in COS-supplemented tilapia were significantly increased compared to those in fish of the control group. There was no significant difference in PA, RB or SOD activity in fish between the G1, G2 and G3 groups (Figure 2a–c). Serum lysozyme activity in tilapia of the G2 and G3 groups was significantly increased compared with that in fish of the control and G1 groups. There was no significant difference in the serum lysozyme activity of fish between the control and G1 groups (Figure 2d). The effect of COSs on the innate response was also studied in-depth at the molecular level by determining the expression of cytokine genes, including *TNF-α*, *IL-1β* and *IL-8*, in the head kidney and spleen. The expression of *TNF-a* and *IL-1β* in the head kidney and spleen of COS-supplemented fish was obviously increased compared to that in control group fish, respectively (Figure 3a,e). The expression of *IL-1β* and *IL-8* in the head kidney and *TNF-α* in the spleen of tilapia in the G2 and G3 groups were significantly increased compared to those in fish in the control and G1 groups (Figure 3b–d). The expression of *IL-8* in the spleen of tilapia in G2 and G3 was significantly increased compared to that in fish in the control group (Figure 3f). In summary, increases in immune parameters and cytokine genes were revealed in tilapia from the G2 and G3 groups, suggesting that fish fed diets containing 0.8% and 1.2% COSs exhibited immunomodulatory functions. Moreover, there were no significant differences in immune parameters or cytokine gene expression between G2 and G3, suggesting that supplementation with COSs at 0.8% was sufficient to enhance innate immunity.

### 3.4. COS Supplementation Enhances Defense against S. Iniae Infection in Tilapia

The significant increase in innate immune parameters in COS-supplemented fish prompted us to evaluate the efficacy of dietary COS supplementation on disease resistance by recording the cumulative mortality of fish after challenge with the *S. iniae* pathogen. The cumulative mortality was maintained at 3.3 ± 5.7% for the control group injected with saline buffer 3 days post-infection. Conversely, the cumulative mortality in the control group injected with *S. iniae* was obviously raised during the first 3 days postinfection and then maintained at 60 ± 10% until 7 days postinfection. Although the cumulative mortality in the G1 group injected with *S. iniae* was slightly lower than that in the control group injected with *S. iniae*, there was no significant difference in cumulative mortality between the groups. Notably, the cumulative mortality rates were 36.7% ± 5.8% and 33.3% ± 5.8% in tilapia from the G2 and G3 groups, respectively, which were significantly lower than those in fish from the control group injected with *S. iniae*. There was no significant difference in cumulative mortality between the G2 and G3 groups (Figure 4). These results suggested that fish fed a COS-supplemented diet had disease resistance against *S. iniae* infection, and a dose of 0.8% COS was sufficient to acquire disease resistance.

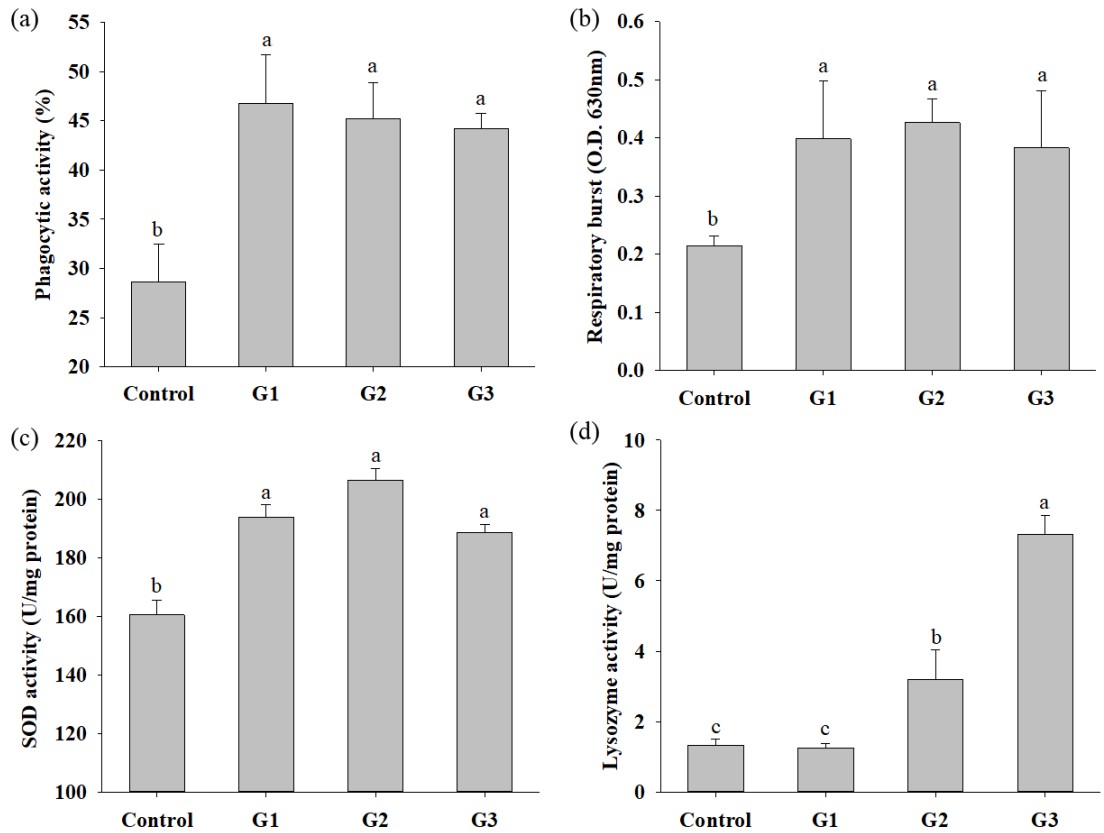

**Figure 2.** Innate immune parameters of Nile tilapia after feeding control diet and control diet containing 4 (G1), 8 (G2) or 12 (G3) g/kg COS for 8 weeks: (**a**) Phagocytic activity; (**b**) Respiratory burst activity; (**c**) SOD activity; (**d**) Lysozyme activity. Bars with different superscripts are significantly different (*p* < 0.05).

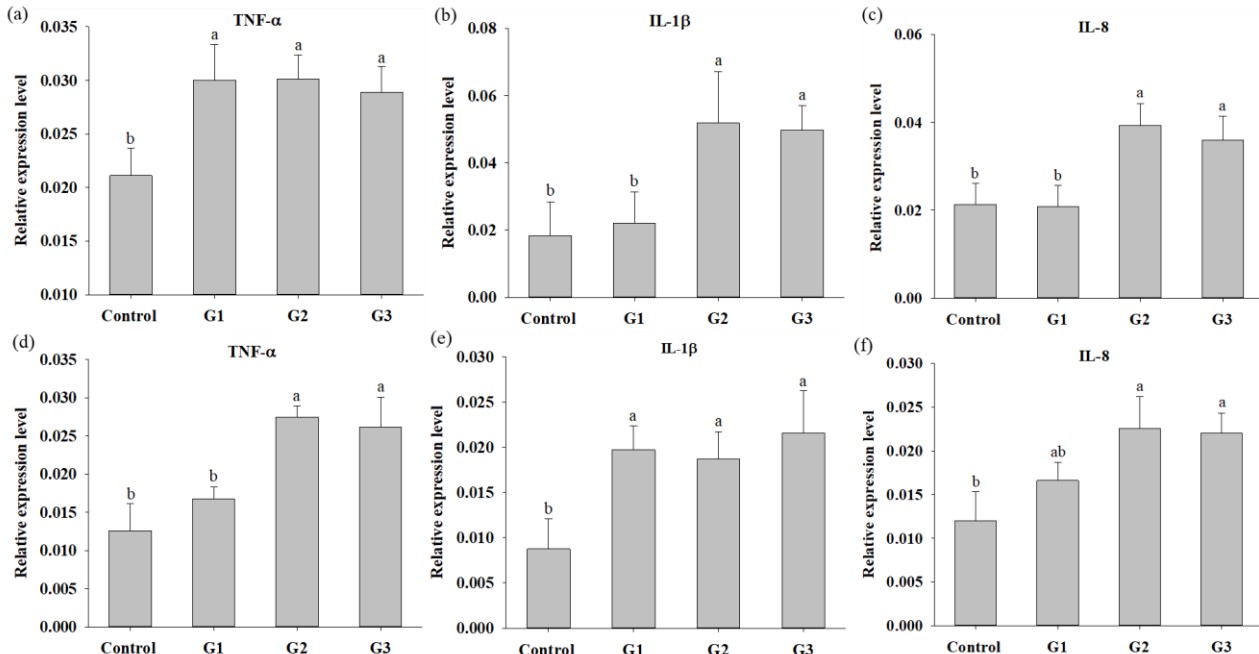

**Figure 3.** Quantitative PCR analysis of cytokine genes expression in Nile tilapia fed with control diet and control diet containing 4 (G1), 8 (G2) and 12 (G3) g/kg COSs for 8 weeks. Expression of *TNF-α*, *IL-1β* and *IL-15* in head kidney: (**a**–**c**) and in spleen; (**d**–**f**). Bars with different superscripts are significantly different (*p* < 0.05).

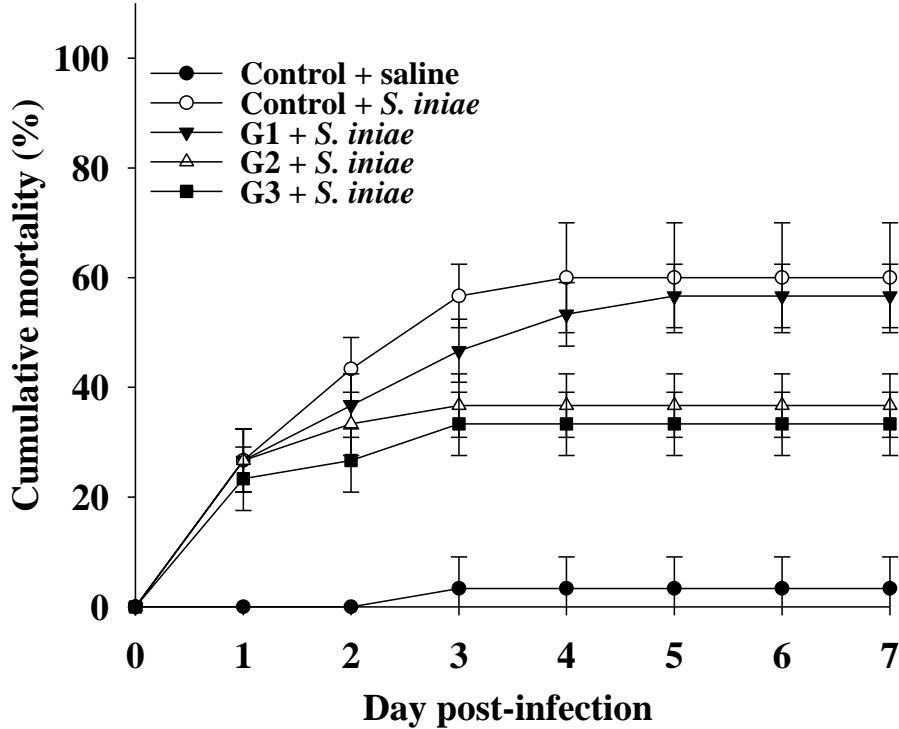

**Figure 4.** Cumulative mortality of Nile tilapia challenged with *S. iniae* after feeding basal diet (control) and basal diet containing 0.4% (G1), 0.8% (G2) or 1.2% (G3) of COS for 8 weeks. The mortality in the G2 and G3 groups were significantly lower than that in the control and G1 group based on the Kaplan–Meier method ($p < 0.05$).

## 4. Discussion

A number of prebiotics, including chitooligosaccharides (COSs), have been widely studied in aquaculture to evaluate their biological functions in aquatic animals. The present study showed that dietary supplementation with COSs improved WG, FE and SGR in Nile tilapia after 8 weeks of feeding. Similar results were also reported in *Trachinotus ovatus* [15], striped catfish (*Pangasianodon hypophthalmus*) [14], Nile tilapia (*O. niloticus*) [25] and tiger puffer (*Takifugu rubripes*) [26]. Recent studies have documented that oligosaccharide prebiotics exhibit the ability to change intestinal morphology, such as increasing villus height, and result in increased digestive enzyme activities in fish, including red drum (*Sciaenops ocellatus*), thinlip gray mullet (*Liza ramada*) and striped catfish (*P. hypophthalmus*) [27–29]. The increase in diverse digestive enzyme activities could contribute to growth performance in fish by enhancing nutrient utilization. In the present study, there was a significant increase in WG, FE and SGR in the G1 group, suggesting that 0.4% COSs is sufficient to promote growth performance. However, only protease activity was significantly increased in fish supplemented with 0.4% COSs. An obvious elevation in digestive enzyme activities, including protease, amylase, cellulase and lipase activities, was shown in fish supplemented with 0.8% COSs. These results suggest that 0.8% COSs may be required to significantly change morphological characteristics and evoke digestive enzyme activities in tilapia intestine, but this hypothesis still needs to be confirmed in further studies. Moreover, nutrition metabolism plays an essential role in supporting and regulating growth. The efficacy of COSs on enhancing growth performance in fish prompted us to investigate the status of nutrition metabolism at the molecular level. The liver is a critical organ responsible for numerous physiological processes, including growth and nutrition metabolism. Hepatic glucokinase (GK) and glucose-6-phosphatase (G6Pase) are responsible for the first and terminal steps of glycolysis and glycogenolysis that catalyze glucose into glucose-6-phosphate and release glucose from glycogen into the circulatory system, respectively, to respond to the energy requirement. Moreover, the hepatic growth hormone (GH) receptor receives

GH ligand from the pituitary gland and stimulates the release of insulin-like growth factor (IGF)-1 from the liver to the blood, which is the main endocrine system responsible for somatic growth. The effect of COSs on glucose metabolism and endocrinal growth has been investigated in livestock and mammals; however, studies have been less investigated in fish. In rats, Jeong et al. demonstrated that COSs are an effective inducer of mitochondrial biogenesis, which is responsible for producing ATPs to meet energy demands by catabolizing glucose [30]. In growing swine, Steele et al. showed that energy intake can increase serum IGF-1 concentrations [31]. Moreover, Tang et al. also reported that dietary supplementation with COSs could improve growth and feed efficiency by increasing hepatic and serum IGF-1 expression levels in early weaned piglets [32]. Similarly, the present study showed that fish supplemented with COSs had evoked expression of hepatic *GK*, *GP*, *GHR* and *IGF-1* genes, suggesting that COSs exhibited a high capacity to promote growth by increasing nutrition utilization, energy intake and secretion of hepatic IGF-1.

Prebiotics modulate immunity against infectious diseases and are one of the major benefits to the host. Although several studies have demonstrated the efficacy of dietary COSs in immune defense against pathogen infection in diverse fish species [13,14,25], the mechanism by which COSs regulate immunity is still not comprehensively understood. Recently, Ouyang et al. demonstrated that the mannose receptor plays an indispensable role in recognizing, binding and mediating COSs into macrophages, thereby activating macrophages to produce cytokines in blunt snout bream (*Megalobrama amblycephala*) [33]. This study potentially elucidated that activation of macrophages and cytokine secretion were important indicators of the immunomodulatory function of COSs. The present study showed that the cumulative mortality of tilapia challenged with *Streptococcus iniae* significantly decreased in fish of the G2 and G3 groups, suggesting that fish fed diets containing 0.8% and 1.2% COSs have enhanced disease resistance. This result prompted us to investigate the efficacy of COSs in modulating innate immune responses at the biochemical and molecular levels. The head kidney and spleen are the most important major immune competent organs in teleosts. Head kidney and splenic lymphocytes, neutrophils or macrophages act as the first defense against pathogenic invasion by recognizing invaded pathogens and thereby engulf invading pathogens by phagocytosis. During the phagocytosis period, respiratory burst (RB) activity, which indicates bactericidal reactive oxygen species (ROS) released from leukocytes, is evoked to kill engulfed pathogens. Thus, phagocytosis and RB of leukocytes are critical bioindicators and are commonly used to evaluate the status of immunity in teleosts. Serum lysozyme, which can destroy invaded pathogens by hydrolyzing peptidoglycan of the bacterial cell wall, is an important component of the innate immune system. Moreover, a recent review reported that lysozyme-mediated bacterial lyses, such as lipopolysaccharide or peptidoglycan fragments, can modulate innate immunity by activating pattern-recognition receptors or complement systems [34]. Thus, serum lysozyme has been used as an indicator of innate immunity to evaluate the health status of teleost fish [35]. The present study showed a significant increase in phagocytic activity (PA) and RB activity in leukocytes of the head kidney and serum lysozyme activity in tilapia of the G2 and G3 groups compared to tilapia of the control group, suggesting that dietary supplementation with diets containing 0.8% and 1.2% COSs can confer immunomodulatory functions in fish. Severe oxidative stress results from excessive ROS production, such as superoxide, hydroxyl, and hydrogen peroxide, which may cause DNA damage, protein denaturation and cell injuries. Superoxide dismutase (SOD) is a first-line antioxidant enzyme that relieves oxidative stress by converting superoxide into hydrogen peroxide, which is thereby decomposed into water by catalase. Thus, SOD activity has been considered a critical indicator of cell protection against oxidative stress. The present study showed a significant increase in SOD activity in the head kidney of COS-supplemented tilapia, suggesting that COSs modulate the protective defense response to the production of ROS during phagocytosis.

Cytokines are important effectors that mediate the innate immune system to act as the first line of host defense against pathogen infection. Tumor necrosis factor (TNF)-α and

interleukin (IL)-1β are the main proinflammatory cytokines implicated in the early stage of pathogen infection. Both are secreted from macrophages and monocytes in response to pathogen infection and then activate the innate immune system by mediating the recruitment and activation of circulating phagocytic cells. IL-8 is a chemoattractant cytokine produced by macrophages and other cell types, such as epithelial cells, that recruits neutrophils and plays an important role in host defense against bacterial infections. The present study showed significantly higher expression of *TNF-α*, *IL-1β* and *IL-8* in the head kidney and spleen of tilapia in the G2 and G3 groups, suggesting that fish fed a diet with 0.8% COS can enhance cytokine production to assist immune defense against pathogen infections. Regarding the dose of COSs, several studies have suggested that appropriate dietary COS supplementation could improve innate immunity against pathogen infections in fish. For example, tilapia supplemented with 0.4% COSs could significantly upregulate the mRNA expression of *IL-1β* and strengthen disease resistance to *Aeromonas hydrophila* [25]; dietary supplementation with 100–200 COSs mg/kg feed could significantly improve immunity against *Edwardsiella ictaluri* in striped catfish (*Pangasianodon hypophthalmus*) [14]; olive flounder (*Paralichthys olivaceus*) supplemented with 1% COSs showed significantly enhanced nonspecific immune response and improved survival rates after challenge with *E. tarda* [13]. Reviews have reported that the biological activity of COSs is strongly correlated with their molecular weight, degree of polymerization, water solubility and applied dose [6,7]. Moreover, Wu et al. demonstrated that an inappropriate molecular weight of COSs or treatment time may cause apoptotic damage in head kidney macrophages in blunt snout bream (*Megalobrama amblycephala*) [36]. These reports elucidated that the physicochemical properties of COSs and treatment conditions are critical parameters for revealing the biological activity of COSs. The optimal dose of COS to reveal immunomodulatory function and disease resistance in the present study was 0.8%. The difference in concentration at the optimal COS dose between the present study and other reports may have resulted from the different molecular weights of COS, treatment times, fish species and other parameters. In summary, the present study showed that the increase in resistance against *S. iniae* by administrating 0.8% COS could be explained by enhanced innate immune responses.

## 5. Conclusions

In conclusion, the present study demonstrated that tilapia fed a diet containing 0.4% COSs exhibited not only increased expression of indicator genes associated with glucose metabolism and growth, but also effectively enhanced growth performance and feed efficiency. Dietary supplementation of COS with 0.8% could improve innate immunity and disease resistance against *S. iniae* infection. The present study showed that COS prebiotics could be effective biologics in improving growth performance, feed utilization and immune defense against pathogen infection in aquaculture.

**Author Contributions:** Conceptualization, S.-Y.H. and C.-H.L.; Data curation, N.; Methodology, N. and I.M.M.; Supervision, S.-Y.H. and C.-H.L.; Writing—original draft, C.-H.L.; Writing—review and editing, S.-Y.H. All authors have read and agreed to the published version of the manuscript.

**Funding:** This study was supported by a grant from the Ministry of Science and Technology (MOST) 108-2313-B-020-005-MY3, Taiwan.

**Institutional Review Board Statement:** The study was conducted according to the guidelines of the Declaration of Helsinki and approved by the Institutional Animal Care and Use Committee (IACUC) of National Pingtung University of Science and Technology (NPUST), Taiwan (approval number: NPUST-107-063).

**Data Availability Statement:** The data presented in this study are available in the article.

**Acknowledgments:** This study was also financially supported by the Research Center for Animal Biologics, from The Featured Areas Research Center Program within the framework of the Higher Education Sprout Project by the Ministry of Education and the National Science and Technology Council (MOST 111-2634-F-020-001-), Taiwan, R.O.C.

**Conflicts of Interest:** The authors declare no conflict of interest.

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
