# Peer review of "Dietary Supplementation with Prebiotic Chitooligosaccharides Enhances the Growth Performance, Innate Immunity and Disease Resistance of Nile Tilapia (Oreochromis niloticus)"

_fishes, doi:10.3390/fishes7060313_

Round 1
Reviewer 1 Report
No comments. Very good work.
Author Response
We thank the Reviewer for the comments. The Reviewers’ comments have been helpful in improving the quality of our manuscript.
Reviewer 2 Report
Polysaccharides/oligosaccharides may contain immunomodulating functions to improve host defence against infective diseases. In this study, chitooligosaccharide supplementation in the feed of Nile tilapia resulted in enhanced growth, enhanced innate immunity and enhanced disease resistance.
The study seems well performed and the results are well presented in the manuscript.
Minor points:
Line 74: and the lower molecular weight of COSs are; change to: and the lower molecular weight of COSs…
L 94: such weight gain; change to: such as weight gain
L 154: PBS; abbreviations should be defined the first time they are mentioned
L 174: PBS buffer; change to: PBS
L 194: potassium phosphate (PBS) buffer (pH 7.4) ??
L 214: Total RNA ….were extracted; change to: Total RNA… was extracted
Table 2: Tumor necrosis factor (TNF-a); change to: Tumor necrosis factor a (TNF-a);
L 319: recoding; change to: recording
L 339: A number….have been; change to: A number…. Has been
L 391: …could have enhanced disease resistance; change to: have enhanced disease resistance
L 423: Both effects are secreted; change to: Both are secreted
L 436: Edwardsiella ictalurid; change to: Edwardsiella ictaluri
Author Response
We thank the Reviewer for the comments. The Reviewers’ comments have been helpful in improving the quality of our manuscript
1. Reviewer’s comment:
Line 74: and the lower molecular weight of COSs are; change to: and the lower molecular weight of COSs…
Authors’ response
In accordance with the Reviewer’s comment, the errors have been corrected in revised manuscript (line 74).
2. Review’s comments:
L 94: such weight gain; change to: such as weight gain
Authors’ response
In accordance with the Reviewer’s comment, the errors have been corrected in revised manuscript (line 94).
3. Review’s comments:
L 154: PBS; abbreviations should be defined the first time they are mentioned
Authors’ response
In accordance with the Reviewer’s comment, the full name of PBS has been added in revised manuscript (line 155).
4. Review’s comments:
L 174: PBS buffer; change to: PBS
Authors’ response
In accordance with the Reviewer’s comment, the errors have been corrected in revised manuscript (line 175).
5. Review’s comments:
L 194: potassium phosphate (PBS) buffer (pH 7.4) ??
Authors’ response
In accordance with the Reviewer’s comment, the errors have been corrected in revised manuscript (line 193).
6. Review’s comments:
L 214: Total RNA ….were extracted; change to: Total RNA… was extracted
Authors’ response
In accordance with the Reviewer’s comment, the errors have been corrected in revised manuscript (line 212).
7. Review’s comments:
Table 2: Tumor necrosis factor (TNF-a); change to: Tumor necrosis factor a (TNF-a);
Authors’ response
In accordance with the Reviewer’s comment, the errors have been corrected in revised manuscript (Table 2).
8. Review’s comments:
L 319: recoding; change to: recording
Authors’ response
In accordance with the Reviewer’s comment, the errors have been corrected in revised manuscript (line 318).
9. Review’s comments:
L 339: A number….have been; change to: A number…. has been
Authors’ response
In accordance with the Reviewer’s comment, the errors have been corrected in revised manuscript (line 338).
10. Review’s comments:
L 391: …could have enhanced disease resistance; change to: have enhanced disease resistance
Authors’ response
In accordance with the Reviewer’s comment, the errors have been corrected in revised manuscript (line 390).
11. Review’s comments:
L 423: Both effects are secreted; change to: Both are secreted
Authors’ response
In accordance with the Reviewer’s comment, the errors have been corrected in revised manuscript (line 422).
12. Review’s comments:
L 436: Edwardsiella ictalurid; change to: Edwardsiella ictaluri
Authors’ response
In accordance with the Reviewer’s comment, the errors have been corrected in revised manuscript (line 435).
Reviewer 3 Report
This is a very well written and interesting paper, the topic is very attractive in the framework of the OneHealth approach and in order to find new indicators of fish welfare, the experimental design is appropriate, a lot of parameters have been considered in a multidisciplinary approach, the results are clearly presented and discussed. I have very few and minor comments:
Keywords: Use the capital letter for Oreochromis
Lines 50-53: please, reformulate the sentence because it is not correct in English
Line 100: use the full name, because the species was never cited before
Line 113: Correct "Trial"
Table 1: there isn't correspondence between the quotes in the table and in the caption
Line 259: did you mean "metabolism"?
Author Response
We thank the Reviewer for the comments. The Reviewers’ comments have been helpful in improving the quality of our manuscript.
1. Review’s comments:
Keywords: Use the capital letter for Oreochromis
Authors’ response
In accordance with the Reviewer’s comment, the errors have been corrected in revised manuscript (line 31).
2. Review’s comments:
Lines 50-53: please, reformulate the sentence because it is not correct in English
Authors’ response
In accordance with the Reviewer’s comment, the sentence have been corrected in revised manuscript (line 52-54).
3. Review’s comments:
Line 100: use the full name, because the species was never cited before
Authors’ response
In accordance with the Reviewer’s comment, the errors have been corrected in revised manuscript (line 100).
4. Review’s comments:
Line 113: Correct "Trial"
Authors’ response
In accordance with the Reviewer’s comment, the errors have been corrected in revised manuscript (line 114).
5. Review’s comments:
Table 1: there isn't correspondence between the quotes in the table and in the caption
Authors’ response
In accordance with the Reviewer’s comment, the errors have been corrected in revised manuscript (Table 1).
6. Review’s comments:
Line 259: did you mean "metabolism"?
Authors’ response
The subtitle has been corrected to fit the meaning of content in revised manuscript (line 255).